# Paracrine Effects of Renal Proximal Tubular Epithelial Cells on Podocyte Injury under Hypoxic Conditions Are Mediated by Arginase-II and TGF-β1

**DOI:** 10.3390/ijms24043587

**Published:** 2023-02-10

**Authors:** Yiqiong Ma, Duilio Michele Potenza, Guillaume Ajalbert, Andrea Brenna, Cui Zhu, Xiu-Fen Ming, Zhihong Yang

**Affiliations:** Cardiovascular & Aging Research, Department of Endocrinology, Metabolism, Cardiovascular System, Faculty of Science and Medicine, University of Fribourg, CH-1700 Fribourg, Switzerland

**Keywords:** arginase, hypoxia, kidney, podocyte, TGF-β1, tubular epithelial cells

## Abstract

Hypoxia is an important risk for renal disease. The mitochondrial enzyme arginase-II (Arg-II) is expressed and/or induced by hypoxia in proximal tubular epithelial cells (PTECs) and in podocytes, leading to cellular damage. Because PTECs are vulnerable to hypoxia and located in proximity to podocytes, we examined the role of Arg-II in the crosstalk of PTECs under hypoxic conditions with podocytes. A human PTEC cell line (HK2) and a human podocyte cell line (AB8/13) were cultured. *Arg-ii* gene was ablated by CRISPR/Case9 in both cell types. HK2 cells were exposed to normoxia (21% O_2_) or hypoxia (1% O_2_) for 48 h. Conditioned medium (CM) was collected and transferred to the podocytes. Podocyte injuries were then analyzed. Hypoxic (not normoxic) HK2-CM caused cytoskeletal derangement, cell apoptosis, and increased Arg-II levels in differentiated podocytes. These effects were absent when *arg-ii* in HK2 was ablated. The detrimental effects of the hypoxic HK2-CM were prevented by TGF-β1 type-I receptor blocker SB431542. Indeed, TGF-β1 levels in hypoxic HK2-CM (but not *arg-ii*^−/−^-HK2-CM) were increased. Furthermore, the detrimental effects of TGF-β1 on podocytes were prevented in *arg-ii*^−/−^-podocytes. This study demonstrates crosstalk between PTECs and podocytes through the Arg-II-TGF-β1 cascade, which may contribute to hypoxia-induced podocyte damage.

## 1. Introduction

Kidney hypoxia plays an important role in renal damage, leading to acute and chronic kidney diseases [1,2]. To maintain electrolyte and fluid homeostasis, an effective glomerular filtration rate (GFR) and high metabolic activities of proximal tubular cells are necessary for secreting and reabsorbing metabolites and water. To accomplish these physiological functions, a particularly high oxygen supply to the kidney is essential. Hence, the kidney, among others, is one of the most vulnerable organs to hypoxic insults [3]. Renal proximal tubular epithelial cells (PTECs) are highly active in the metabolic process and reveal high oxygen demand, and are sensitive and vulnerable to hypoxia [4]. It has been reported that hypoxic insult induces a series of pathological processes causing PTEC apoptosis [5,6]. PTECs are also able to produce cytokines such as TGF-β1, participating in renal inflammation and fibrosis in response to insults, including hypoxia [7]. It is important to recognize that TGF-β1 exerts multiple functions participating in several cellular processes, leading to organ damage. The cytokine is involved in the epithelial–mesenchymal transition (EMT), downregulation of cellular tight junctions [8], remodeling of extracellular matrix (ECM), favoring collagen production and/or degradation [9,10], cell proliferation, and cell survival [11]. Moreover, kidney hypoxia causes albuminuria and reduces GFR, which is accounted for at least in part by podocyte dysfunction [12]. Podocytes, anchored to the glomerular basement membrane, are the terminally differentiated specialized epithelial cells with abundant foot processes that are the important components of the glomerular filtration barrier [13]. Compelling evidence shows that podocyte injury is considered a crucial event in the onset and development of kidney diseases [14,15], and hypoxia causes podocyte injury resulting in slit-diaphragm dysfunction, foot process effacement, and cytoskeleton derangement linked to the accumulation of hypoxia-inducible factors (HIFs) [12,16]. Our recent study demonstrates that the mitochondrial enzyme arginase-II (Arg-II) can be upregulated or induced in both PTECs and podocytes under hypoxic conditions and contributes to kidney damage involving mitochondrial oxidative stress [6,17].

*Arg-ii* has been identified to be one of the HIF target genes [17,18,19,20,21,22,23]. In mammals, there are two isoforms of arginase. Arg-I is highly expressed in the liver and located in the cytosol, whereas Arg-II is located in the mitochondria and highly expressed in the kidney, especially the S3 PTECs [24,25]. Both isoenzymes catalyze the hydrolysis of L-arginine to L-ornithine and urea [26]. Although emerging evidence demonstrates an important role of Arg-II in renal pathophysiology, including hypoxia conditions [6,17,27], the exact mechanisms remain largely unknown.

It is reported that glomerulotubular balance (GTB) and tubuloglomerular feedback (TGF) influence the development of various kidney diseases [28,29]. Given that PTECs and glomeruli are located in proximity, we hypothesize that Arg-II in PTECs may play a role in podocyte injury via a paracrine mechanism under hypoxic conditions.

## 2. Results

### 2.1. Arg-II Deficiency in Human HK2 Epithelial Cells Prevents Paracrine Effects on Podocyte Cytoskeleton Filament Derangement

Generation of the *arg-ii* knockout human proximal epithelial cell line, *arg-ii*^−/−^-HK2, by CRISPR/Cas9 was confirmed by immunoblotting (Figure 1A). Conditioned media (CM) from *wt*-HK2-cells and *arg-ii*^−/−^-HK2 exposed to normoxia or hypoxia for 48 h were then collected and transferred to the differentiated podocytes (Figure 1B). Differentiated podocytes treated with normoxic *wt*-HK2-CM for 24 h displayed long, parallel, and organized cytoskeleton filament fibers (Figure 1C), whereas podocytes treated with hypoxic *wt*-HK2-CM for the same time period exhibited significant cytoskeleton filament reorganization. The cytoskeleton fiber organization pattern became shortened and disrupted, and the percentage of cells with cytoskeleton fiber disruption was increased (Figure 1C,D). This change in podocyte cytoskeleton filament reorganization or disruption was prevented when the *arg-ii* gene was knocked out in the HK2 cells (Figure 1C,D). β-catenin, which has been reported to be closely associated with cellular cytoskeleton actin organization [30,31], was significantly decreased in the podocytes treated with hypoxic *wt*-HK2-CM, which was prevented with *arg-ii*^−/−^
*in* HK2 cells (Figure 1E,F).

### 2.2. Arg-II Deficiency in Human HK2 Epithelial Cells Prevents Paracrine Effects on Podocyte Apoptosis

Further experiments showed that hypoxic *wt*-HK2-CM but not *arg-ii*^−/−^-HK2-CM, causes cellular rounding in the morphology of podocytes in culture (Figure 2A). In addition, TUNEL staining reveals podocyte apoptosis induced by hypoxic *wt*-HK2-CM, which was not observed with *arg-ii*^−/−^-HK2-CM (Figure 2B). In accordance, the levels of cleaved-caspase-3 associated with cellular apoptosis were increased in podocytes treated with hypoxic *wt*-HK2-CM but not *arg-ii*^−/−^-HK2-CM (Figure 2C). These results suggest a paracrine factor(s) released from HK2 cells, causing podocyte apoptosis, which is dependent on Arg-II in the HK2 epithelial cells.

### 2.3. Hypoxic HK2-CM Enhances Arg-II Levels in Podocytes through TGF-β1

Interestingly, differentiated human podocytes exposed to hypoxic but not normoxic *wt*-HK2-CM over 24 h exhibit enhanced Arg-II protein levels (Figure 3A,B). This effect was not observed with the hypoxic *arg-ii*^−/−^-HK2-CM (Figure 3A,B). The protein levels of podocin, located in the podocyte foot process, are not significantly altered by HK2-CM in the different groups (Figure 3A,C). No changes in the protein levels of TGF-β1 were observed in the different groups of podocytes (Figure 3A,D), demonstrating that HK2 cells under hypoxic conditions exert paracrine effects on podocytes to upregulate Arg-II in these latter cells. As shown in Figure 3E,F, the increase in Arg-II expression in podocytes induced by *wt*-HK2-CM was inhibited in the presence of TGF-β1 receptor-I inhibitor SB431542 (10 μmol/L for 1 h). Moreover, ELISA confirmed elevated TGF-β1 levels hypoxic *wt*-HK2-CM but not in *arg-ii*^−/−^-HK2-CM (Figure 3G), suggesting that TGF-β1 could be the paracrine mediator released from HK2 cells under hypoxic conditions. In support of this notion, TGF-β1 receptor-I inhibitor SB431542 was also able to prevent cytoskeleton filament derangement induced by hypoxic *wt*-HK2-CM in podocytes (Figure 4). The results demonstrate that hypoxia induces the release of TGF-β1 from HK2 cells, which causes Arg-II upregulation and cell injury in podocytes.

### 2.4. Role of Arg-II in Podocytes in TGF-β1-Induced Podocytes Damage

Since Arg-II is upregulated in podocytes by hypoxic *wt*-HK2-CM but not *arg-ii*^−/−^-HK2-CM, and TGF-β1 is the paracrine factor released from the hypoxic HK2-cells, we investigated the hypothesis that Arg-II upregulation in podocytes may play a role in TGF-β1-mediated podocyte injury. To test this hypothesis, *arg-ii*^−/−^-podocytes were generated by CRISPR/Cas9 technology. The effects of TGF-β1 on *wt* and *arg-ii*^−/−^ podocytes were examined after stimulation of podocytes with TGF-β1 (10 ng/mL, 48 h) and cellular injury was analyzed. The results show that treatment of TGF-β1 indeed induced cytoskeleton filament derangement in podocytes, which was prevented by *arg-ii* knockout (Figure 5A,B). Furthermore, Arg-II protein levels were enhanced by TGF-β1 in the *wt*-podocytes, which is associated with elevated cleaved-caspase-3 protein levels (Figure 5C–E). This effect of TGF-β1 was inhibited in the *arg-ii*^−/−^-podocytes (Figure 5C–E), demonstrating that TGF-β1 induces cytoskeleton derangement and podocyte injury through Arg-II in the podocytes.

## 3. Discussion

In the kidney, tubules and glomeruli have intrinsic mechanisms to regulate each other, which is called glomerulotubular balance and tubuloglomerular feedback to keep the internal balance of renal hemodynamics [32]. In recent years, the roles of tubular epithelial cell injury in triggering and progression of renal diseases have attracted much attention. As the main part of tubules, the PTECs play an important role in the pathophysiological process of kidney diseases and also during aging [4,25,33,34]. Clinical studies suggest that tubular damage precedes glomerular damage and may interact with glomerular cells contributing to glomerular functional changes, for example, in diabetic nephropathy [35]. The PTECs are capable of producing various inflammatory cytokines or factors, such as IL1β, TGF-β1, etc. [32,36,37]. Recent studies also show that PTECs, upon injury, affect not only the neighbouring cells but also the distal tubule and collecting duct cells through release of exosomes, contributing to tubulointerstitial fibrosis via activation of profibrotic signalling pathways [38]. In line with these findings, our results show that conditioned medium (CM) from human proximal HK2 epithelial cells exposed to hypoxia (but not normoxia) causes derangement of cytoskeletal filaments and cell apoptosis in cultured podocytes. These results imply the paracrine effect of PTECs under hypoxia on podocyte injury. The fact that TGF-β1 concentration is increased in the hypoxic HK2-CM and the podocyte injury induced by hypoxic HK2-CM is prevented by TGF-β type-I receptor antagonist, demonstrates that the secreted TGF-β1 from PTECs mediates the crosstalk between injured PETCs and podocytes under hypoxic conditions. This conclusion is further supported by the finding that HK2-CM does not enhance TGF-β1 levels in podocytes.

The second important finding of our current study is that HK2 epithelial cells produce and release TGF-β1 which acts as a paracrine mediator on podocytes and causes podocyte damage through upregulation of Arg-II in the podocytes. Our previous studies demonstrate that hypoxia upregulates TGF-β1 expression in tubular epithelial cells through the HIF1α-Arg-II-mitochondrial ROS pathway, which causes the collagen gene expression and renal epithelial injury [6]. Up-regulation of Arg-II could also be observed in podocytes under hypoxic conditions, which contributes to podocyte injury [17]. Interestingly, our present study further demonstrates that Arg-II is also upregulated by TGF-β1 released from hypoxic epithelial cells. The detrimental effect exerted by hypoxic epithelial CM or by TGF-β1 is dependent on Arg-II in the podocytes since genetic ablation of *arg-ii* in the podocytes are resistant to TGF-β1-induced damage. These results demonstrate an important role of Arg-II in podocyte injury. How TGFβ1 upregulates Arg-II in podocytes remains to be investigated.

It is of note that crosstalk among podocytes and other cell types in the kidney has been demonstrated in the literature. For example, during CKD progression, vascular endothelial growth factor (VEGF)-A/C and angiopoietin-1 are secreted by podocytes, which results in glomerular endothelial cell dysfunction and albuminuria [39,40,41]. Moreover, podocyte–mesangial cell communication has also been reported. While podocyte injury promotes mesangial cell proliferation, mesangial cell injury also causes foot process fusion and proteinuria [42]. Factors derived from mesangial cells induce epithelial–mesenchymal transition (EMT) of podocytes, leading to glomerular injury [43]. In addition, it has been reported that extracellular vesicles are secreted from high glucose-treated podocytes to induce apoptosis of PTECs [44]. Vice versa, some humoral mediators released from proximal tubules are proposed to cause podocyte dysfunction [45]. However, research into the mechanism and implication of podocyte-PTEC crosstalk in the progression of podocytes and tubulointerstitial injury are not clear or limited. Our present study adds another piece of evidence demonstrating that hypoxic tubular cells play a role in podocyte damage through the paracrine release of TGF-β1, which is dependent on Arg-II.

Although the interesting crosstalk between cultured PETCs and podocytes under hypoxic conditions is identified in the current study, further experiments need to be conducted in animal models to validate the findings in vivo. Especially, PTEC-specific *arg-ii* and/or *tgf-β1* knockout mice and podocyte-specific *arg-ii* knockout mice shall be developed and used to confirm the vital role of Arg-II-TGF-β1 cascade during the crosstalk between PTECs and podocytes. Of course, it would be challenging to develop animal models which have only hypoxic PTECs, while podocytes remain normoxic.

Nevertheless, our present study using in vitro cell culture model suggests a paracrine effect of hypoxic PTECs on podocyte injury. Hypoxic tubular epithelial cells release TGF-β1 through Arg-II. The epithelial-derived TGF-β1 then upregulates Arg-II in the podocytes, leading to podocyte injury (Figure 6). Notably, hypoxia occurs in many situations, e.g., renal ischemia or interstitial vascular rarefaction in diabetes and aging [46], which is closely related to acute and chronic kidney disease development [1,2,47]. The results of our study on the crosstalk between PTECs and podocytes provide important novel pathophysiologic insight into the development of kidney diseases under many pathological conditions. It also suggests that targeting *arg-ii* could be a promising therapeutic approach for the treatment of kidney diseases.

## 4. Materials and Methods

### 4.1. Reagents

Reagents were purchased or obtained from the following sources: rabbit antibody against Arg-II (#55003), cleaved-caspase 3 (#9664), and β-catenin (#8480) were from Cell Signaling Technology (Danvers, MA, USA); rabbit antibody against podocin (PA5-37284) was from Invitrogen/Thermo Fisher Scientific (Waltham, MA USA); rabbit antibody against TGF-β1 (ab215715) was from Abcam (Cambridge, United Kingdom); mouse antibody against GAPDH (10R-G109a), was from Fitzgerald (Acton, MA USA); IRDye 800-conjugated affinity purified goat anti-rabbit IgG F(c) was purchased from LI-COR Biosciences (Lincoln, NE, USA); goat anti-mouse IgG (H + L) secondary antibody Alexa Fluor^®^ 680 conjugate was from Invitrogen/Thermo Fisher Scientific (Waltham, MA, USA). These antibodies and their dilutions used for immunoblotting are presented in Table 1.

### 4.2. Cell Culture

Conditionally immortalized human podocytes (AB8/13) were kindly provided by Andreas Kistler (University of Zürich) [48] and cultured in RPMI 1640 medium (BioConcept) containing 10% fetal bovine serum (FBS), 100 U/mL penicillin and 100 μg/mL streptomycin (Gibco), 1× insulin, transferrin, and selenium (ITS) in a 33 °C incubator with 5% CO_2_ for proliferation. When the podocytes were about 50% confluent, the cells were moved from 33 °C to 37 °C for differentiation for 10–14 days. The HK-2 cells (a human proximal tubular epithelial cell line, PTECs) were purchased from American Type Culture Collection (ATCC, Manassas, VA, United States) and cultured in Dulbecco modified Eagle medium/F12 (DMEM/F12) supplemented with 10% FBS, 100 U/mL penicillin, and 100 μg/mL streptomycin under standard conditions at 37 °C and 5% CO_2_. For staining experiments, cells were cultured on coverslips coated with 1% gelatin. For collecting hypoxic conditioned media (CM), HK2 cells were cultured in Hypoxic Cabinet System for in vitro studies (1% O_2_, The Coy Laboratory Products, Grass Lake, USA) for 48 h. The culture media was changed every other day.

### 4.3. Generation of Arg-Ii Knockout HK2 Cell Line and Podocyte Cell Line by CRISP/Cas9 Technologies

sgRNA targeting human *arg-ii* (the top strand of the sgRNA that recognizes the target DNA region of human *arg-ii*: GGGACTAACCTATCGAGA) was cloned into pSpCas9(BB)-2A-Puro (PX459) V2.0 (Plasmid #62988, Addgene) to generate pSpCas9(BB)-2A-Puro (PX459)-U6/sgRNA-h*arg-ii*. AB8/13 cells (cultured in a 33 °C incubator) and HK2 cells were plated in a 6-cm dish at a density of 1 × 10^6^ cells 24 h before transfection. Transfection of pSpCas9(BB)-2A-Puro (PX459)-U6/sgRNA-h*arg-ii* was performed using Lipofectamine™ 3000 Transfection Reagent (L3000008, Invitrogen™) according to the manufacturer’s protocol. Briefly, per 1 × 10^6^ cells, diluted plasmid DNA (5 µg, diluted with P3000™ Reagent) and diluted Lipofectamine™ 3000 Transfection Reagent were mixed at a 1:1 ratio and incubated at room temperature for 15 min. The DNA-lipid complex was then added to the cells. To select the sgRNA-positive cells, 48 h post-transfection, cells were treated with puromycin (2.5 µg/mL) for 48 h until all the control cells without transfection died. Puromycin-resistant cells were allowed to recover in the medium without puromycin for one week before seeding single cells into a 96-well plate by dilution. Single clones were then expanded and screened for Arg-II by immunoblotting.

### 4.4. Crosstalk between HK2 Cells and Podocytes

For these experiments, HK2 cells were seeded in six-well plates with a density of 2 × 10^5^ cells per well. Before experiments, the cells were serum starved for 24 h. *Wt* and *arg-ii*^−/−^ HK2 cells were exposed to normoxia (21% O_2_) or hypoxia condition (1% O_2_) for 48 h. The CM were collected from the HK2 and then filtered and transferred to cultured podocytes for 24 h. To study whether HK2-CM exerts its effects on podocyte injury through releasing TGF-β1, cultured podocytes were pre-treated with TGF-β receptor 1 inhibitor SB431542 (S1067, Selleck, 10 μmol/L) before incubation with the CM.

### 4.5. Immunoblotting

Cells were lysed on ice with lysis buffer containing 10 mmol/L Tris-HCl (pH 7.4), 0.4% Triton X-100, 10 μg/mL leupeptin, and 0.1 mmol/L phenylmethylsulfonyl fluoride (PMSF), protease inhibitor cocktail (B14002) and phosphatase inhibitor cocktail (B15002; Bio-tool). After frozen and thawed twice with liquid nitrogen, the cell lysates were centrifuged at 12,000 rpm for 20 min at 4 °C. The supernatants were taken and the protein concentration was determined using the Lowry protein assay. Equal amounts (30 μg) of protein were mixed with loading buffer and boiled at 95 °C for 5 min and separated using 10% sodium dodecyl sulfate-polyacrylamide gel electrophoresis (SDS-PAGE), then transferred to an Immobilon-P membrane (Millipore), and the resultant membrane was incubated overnight with the corresponding primary antibody at 4 °C with gentle shaking after being blocked with 5% skimmed milk. The blot was then further incubated with a corresponding anti-mouse (Alexa fluor 680 conjugated) or anti-rabbit (IRDye 800 conjugated) secondary antibody. Signals were visualized using Odyssey Infrared Imaging System (LI-COR Biosciences, Lincoln, Nebraska USA). Quantification of the signals was performed using NIH Image J 1.50i software (National Institutes of Health, USA).

### 4.6. Cytoskeleton Staining

Cells grown on coverslips were fixed with 4% formaldehyde at room temperature for 30 min and washed three times with PBS for 5 min. Then the fixed cells were incubated with 0.4 nmol/mL Atto 488 phalloidin (Sigma-Aldrich, St. Louis, MO, USA) for F-actin staining for 60 min at room temperature and DAPI (4’6-diamidino-2-phenyl-indole dihydrochloride, Invitrogen, 300 nmol/L) was added simultaneously for counterstaining of nuclei. Images were acquired through 63× objectives with Leica TCS SP5 confocal laser microscope. Ten fields of view were randomly selected, and the percentage of podocytes with disrupted actin was quantified.

### 4.7. TUNEL Assay

The detection of apoptotic podocytes was carried out by staining cells on coverslips with the “In Situ Cell Death Detection Kit, TMR red” (TUNEL) (Roche Applied Science, #12156792910, Basel, Switzerland) according to the manufacturer’s instructions. The signals were visualized under Leica TCS SP5 confocal microscope. Quantification of the TUNEL-positive cells was performed using NIH Image J software (version 1.49). 

### 4.8. Enzyme-Linked Immunosorbent Assay (ELISA)

TGF-β1 levels in HK2-CM were measured by ELISA kit (88-8350-22, Invitrogen ThermoFisher, Waltham, MA, USA) according to the manufacturer’s instructions.

### 4.9. Statistics

Data are given as mean ± SD, and all the experiments in the present study were performed independently at least three times. Statistical analysis was performed with the Student’s t-test for unpaired observations or ANOVA with Bonferroni’s post-test, *p* < 0.05 was considered statistically significant.

## Figures and Tables

**Figure 1 ijms-24-03587-f001:**
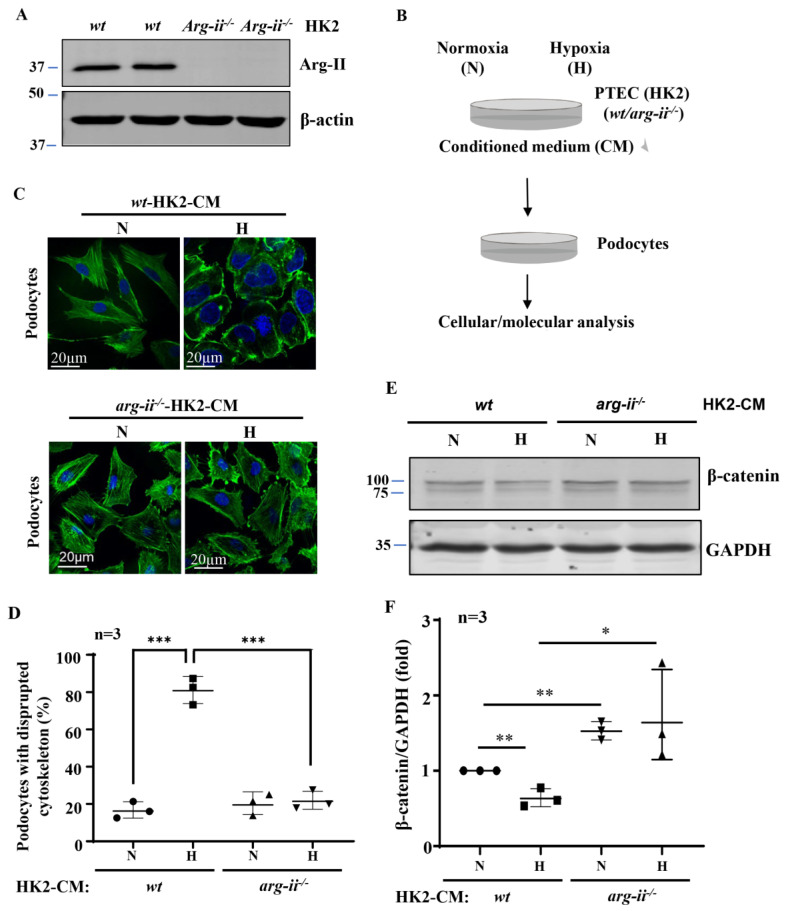
*Arg-ii* deficiency in human renal epithelial cells prevents hypoxia-induced paracrine effects of the cells on podocyte cytoskeleton derangement. (**A**) Immunoblotting revealing *arg-ii* deficiency in *arg-ii*^−/−^ HK2 cells. (**B**) Schematic illustration of the experimental setup to study crosstalk between proximal tubular epithelial cells (PTECs) and podocytes. Human PTECs (HK2-CRISPR-*wt* and -CRISPR-*arg-ii*^−/−^) were exposed to either normoxia (21% O_2_) or hypoxia (1% O_2_) conditions for 48 h. Conditioned medium (CM) was then collected from HK2 cells and transferred to the human differentiated podocyte (AB8/13) for 24 h of incubation. Cell lysates of AB8/13 were then prepared and subjected to downstream analysis. (**C**) Representative images showing phalloidin staining of cytoskeletal actin fibers (green) in human podocytes treated with different HK2-CM as indicated. The nucleus was stained with DAPI (blue). (**D**) The quantification of the podocytes with disrupted actin cytoskeleton (in percentage). (**E**) Immunoblotting analysis of the protein levels of β-catenin; GAPDH serves as the loading control. (**F**) Quantification of the β-catenin signals. ** p* < 0.05, ** *p* < 0.01, *** *p* < 0.001 between the indicated groups. n = 3; CM: conditioned media, N: normoxia, H: hypoxia.

**Figure 2 ijms-24-03587-f002:**
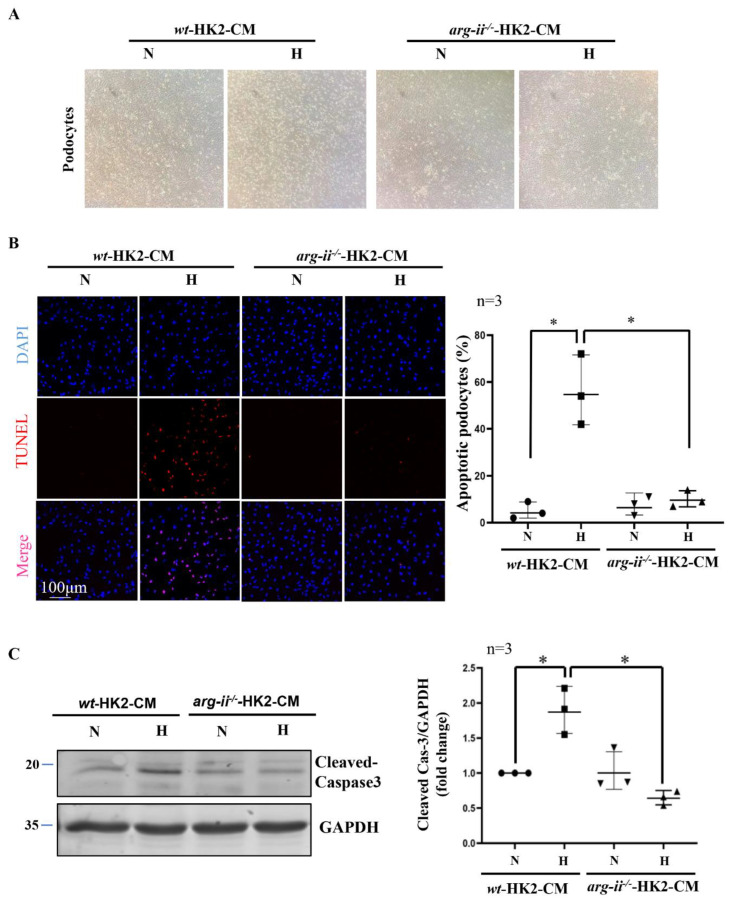
*Arg-ii* deficiency in human renal epithelial cells prevents hypoxia-induced paracrine effects of the cells on podocyte apoptosis. (**A**) Representative images showing cell morphology in different groups of podocytes. (**B**) Representative images of TUNEL staining in human podocytes. Nuclei were stained with DAPI (blue). The quantification of apoptotic podocytes is presented in the graph on the right. (**C**) Cell apoptosis was monitored by immunoblotting analysis of the protein levels of cleaved-caspase 3; GAPDH serves as the loading control. The graph on the right shows the quantification of the signals of cleaved-caspase 3. * *p* < 0.05 between the indicated groups. n = 3; CM: conditioned media, N: normoxia, H: hypoxia.

**Figure 3 ijms-24-03587-f003:**
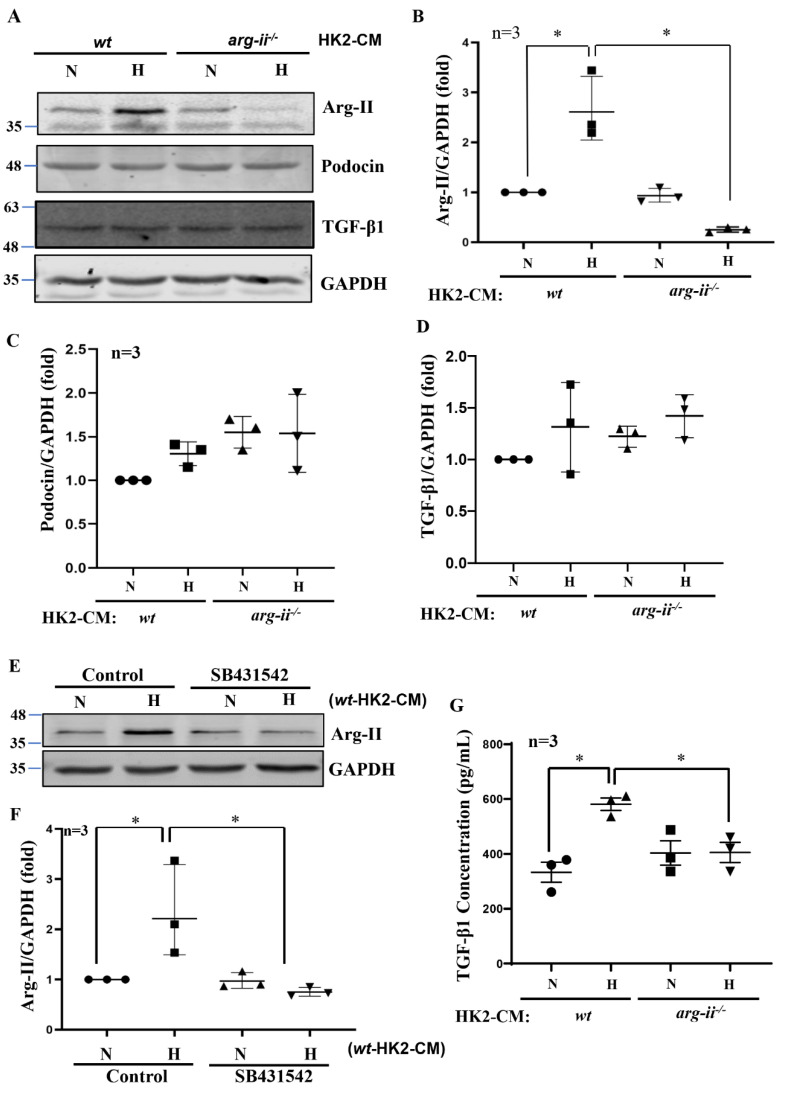
Hypoxic HK2 enhances Arg-II levels in podocytes through TGF-β1. Experiments were performed as described in Figure 1, and podocyte lysates were prepared and subjected to (**A**) Immunoblotting analysis of the protein levels of Arg-II, podocin, and TGF-β1; GAPDH serves as the loading control. The quantification of the signals is presented as graphs in (**B**–**D**). (**E**) Podocytes were treated with *wt*-HK2-CM in the absence or presence of SB431542: a TGF-β1 receptor 1 inhibitor. (**F**) Quantification of Arg-II signal in (**E**). (**G**) Quantification of the concentration of TGF-β1 in HK2-CM measured by ELISA. ** p* < 0.05 between the indicated groups. n = 3; CM: conditioned media, N: normoxia, H: hypoxia.

**Figure 4 ijms-24-03587-f004:**
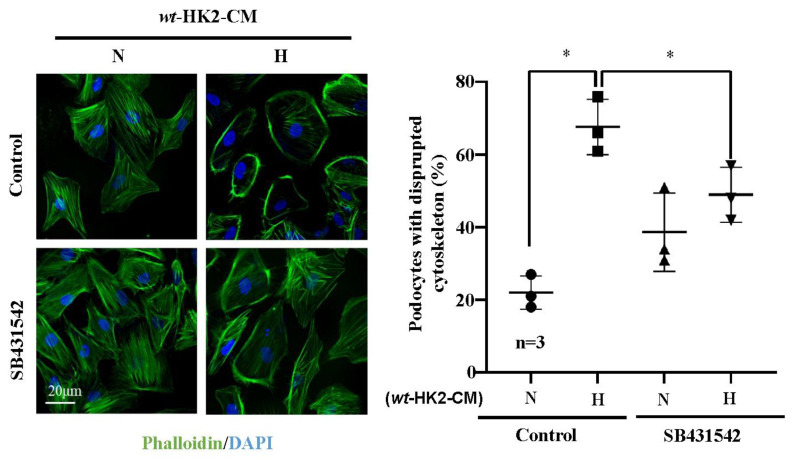
Hypoxic HK2 enhances cytoskeleton filament derangement in podocytes through TGF-β1. Experiments were performed as described in Figure 3E and subjected to Phalloidin staining of cytoskeletal actin fibers (green) in human podocytes. Nuclei were stained with DAPI (blue). Quantification of the podocytes with disrupted actin cytoskeleton (in percentage) is presented in the graph on the right. * *p* < 0.05 between the indicated groups. n = 3; CM: conditioned media, N: normoxia, H: hypoxia. SB431542: TGF-β1 receptor 1 inhibitor.

**Figure 5 ijms-24-03587-f005:**
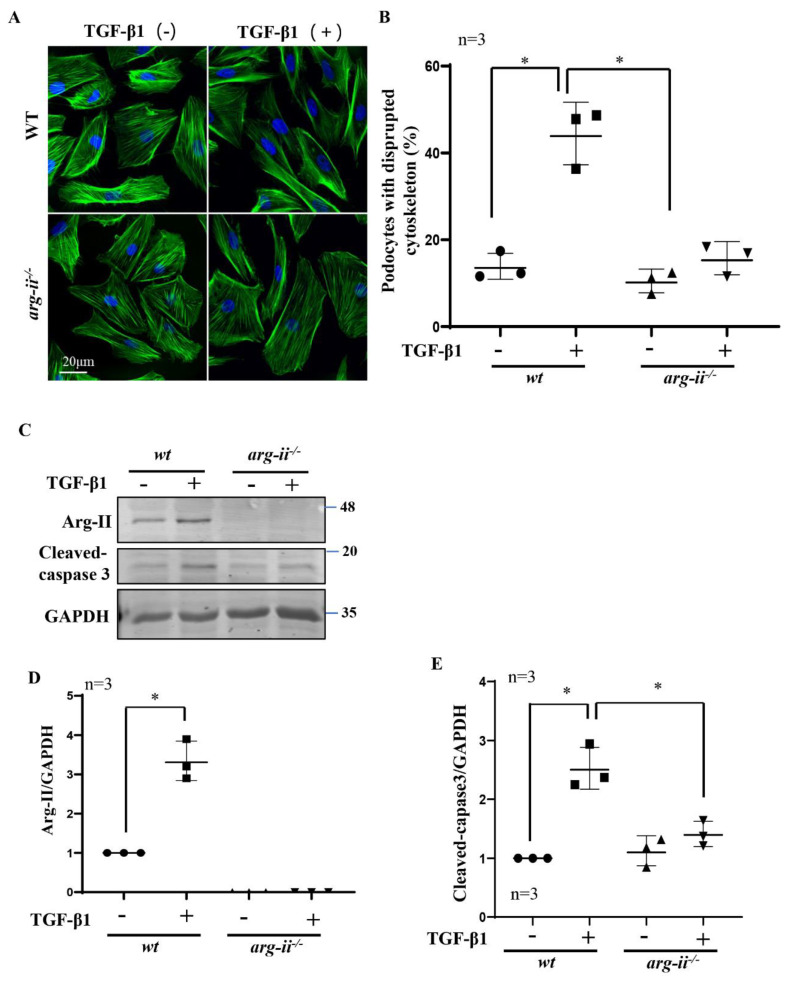
*Arg-ii* knockout in podocytes prevents TGF-β1-induced podocyte cytoskeleton disorganization and apoptosis. Podocytes were treated with or without TGF-β1 (10 ng/mL) for 48 h and subjected then to (**A**) phalloidin staining of cytoskeletal actin fibers (green). The nucleus were stained with DAPI (blue). (**B**) Quantification of podocytes with disrupted actin cytoskeleton is presented in the graph. (**C**) Immunoblotting analysis of the protein levels of Arg-II and cleaved-caspase 3 in different groups of podocytes as indicated; GAPDH serves as the loading control. (**D**,**E**) Graphs showing quantification of the signals of Arg-II and cleaved-caspase 3, respectively. * *p* < 0.05 between the indicated groups. n = 3.

**Figure 6 ijms-24-03587-f006:**
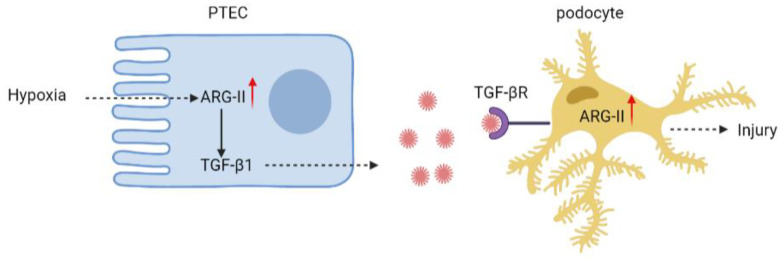
Schematic summary of the current study. Hypoxia increases Arg-II levels leading to enhanced TGF-β1 production and release from PTEC. The paracrine released TGF-β1 from PTECs activates its receptor on podocytes to increase Arg-II, leading to podocyte injury.

**Table 1 ijms-24-03587-t001:** Antibodies and dilutions used for immunoblotting.

Antibody Target	Dilution
Arg-II (Cell Signal, 55003)	WB 1:1000
Cleaved-caspase 3 (Cell Signal, 9664),	WB 1:1000
β-catenin (Cell Signal, 8480)	WB 1:1000
Podocin (PA5-37284, Invitrogen/Thermo Fisher Scientific)	WB 1:500
β-actin (A5441, Sigma-Aldrich)	WB 1:10,000
GAPDH (10R-G109a, Fitzgerald)	WB 1:10,000
TGF-β1 (ab215715, Abcam)	WB 1:1000
IRDye 800-conjugated affinity purified goat anti-rabbit IgG F(c) (LI-COR Biosciences)	WB 1:10,000
Alexa fluor 680-conjugated goat anti-mouse IgG (Invitrogen)	WB 1:10,000

## Data Availability

Data supporting the present study are available from the corresponding author upon reasonable request.

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
