# Peer review of "Paracrine Effects of Renal Proximal Tubular Epithelial Cells on Podocyte Injury under Hypoxic Conditions Are Mediated by Arginase-II and TGF-β1"

_ijms, 2023, doi:10.3390/ijms24043587_

Round 1

Reviewer 1 Report

In this study authors investigated the crosstalk between PECs and podocytes through Arg-II-TGF-β1 cascade by using HK2 epithelial cells and Conditionally immortalized human podocytes (AB8/13). 

The manuscript is interesting and generally well written. However, some points deserve to be improved. Such as:

Introduction: authors should highlight the pleiotropic role of TGFB1. In fact, it is involved in regulating several cellular processes such as Epithelial-mesenchymal transition (EMT) (downregulating tight junctions expression PMID: 26739007), extracellular matrix (ECM) omeostasis and remodelling (favoring collagen production PMID: 32006713 or ECM degradation (PMID: 35131488).

Figure 6: Fig. 6 in the figure must be removed

4.1. Reagents: the antibodies listed in this section should be insert in a table. Moreover, antibodies dilution must be reported 

An accurate revision of formatting is recommended. For example, line 112, graph and GAPDH. Formatting must be uniform through the text

Reviewer 2 Report

This is an interesting and generally well written manuscript highlighting the crosstalk between PECs and podocytes through Arg-II-TGF-β1 cascade.

Only minor points could be improved. In particular: 

Lines 188-192: It deserves to be highlighted that TGFB1 is also an important pro-survival factor that promotes cell proliferation (See PMID: 26708185). This is an important point to highlight since it can further validate the interesting results obtained by the authors 

The molecular weight should be reported in figures showing western blot images 

4.5. Immunoblotting: Dilutions of primary antibodies must be reported  

4.8. Enzyme-Linked Immunosorbent Assay (ELISA): Product code must be reported
